# The Perspectives of Fertility Preservation in Women with Endometrial Cancer

**DOI:** 10.3390/cancers13040602

**Published:** 2021-02-03

**Authors:** Jure Knez, Leyla Al Mahdawi, Iztok Takač, Monika Sobočan

**Affiliations:** 1Department of Gynaecology and Obstetrics, Faculty of Medicine University of Maribor, Taborska ulica 8, 2000 Maribor, Slovenia; iztok.takac@ukc-mb.si (I.T.); monika.sobocan@ukc-mb.si (M.S.); 2Divison for Gynaecology and Perinatology, University Medical Centre Maribor, Ljubljanska ulica 5, 2000 Maribor, Slovenia; Leyla.AlMahdawi@ukc-mb.si

**Keywords:** endometrial cancer, fertility sparing, reproductive function, young women, molecular classification, quality of life

## Abstract

**Simple Summary:**

Endometrial cancer is a common gynecological malignant disease. Its incidence in women of reproductive age in developed countries is increasing. The standard treatment is surgical in the form of hysterectomy and bilateral salpingo-oophorectomy, which has a significant impact on the quality of women’s lives and precludes further fertility. Conservative management to preserve reproductive function and delay final surgery can today be considered in carefully selected women. We analyze the current approaches to select appropriate candidates and current medical regimens for fertility sparing management. We elaborate on the future perspectives of management. With better characterization of the disease and implementation of molecular biomarkers, more women should be able to benefit from conservative approaches to management of endometrial cancer.

**Abstract:**

Endometrial cancer is the most common gynecological cancer in developed countries. The disease is diagnosed with increasing frequency in younger women, commonly also in their reproductive age. The standard treatment of endometrial cancer is surgical in the form of hysterectomy and bilateral salpingo-oophorectomy, and this precludes future fertility in younger women. The current challenge is to identify the group of women with endometrial cancer and low-risk features that would benefit from more conservative treatment options. More focus in management needs to be aimed towards the preservation of quality of life, without jeopardizing oncological outcomes. In this review, we analyze the current approaches to identification of women for conservative management and evaluate the success of different medical options for treatment and surgical techniques that are fertility sparing. We also elaborate on the future perspectives, focusing on the incorporation of molecular characterization of endometrial cancer to fertility preservation algorithms. Future studies should focus specifically on identifying reliable clinical and molecular predictive markers in this group of young women. With improved knowledge and better risk assessment, the precision medicine is the path towards improved understanding of the disease and possibly widening the group of women that could benefit from treatment methods preserving their fertility.

## 1. Introduction

Endometrial cancer is the most common gynecological malignancy in developed countries today. It is the fourth most common cancer in women in the United Kingdom and United States [1,2]. There were over 380,000 new cases diagnosed worldwide in 2018 [3]. It is most commonly a disease of postmenopausal women, but the incidence in younger women of reproductive age is also increasing. Women younger than 40 years represent up to 5% of endometrial cancer cases and around 20% of cases are diagnosed before menopause [4]. This is partly related to increasing global epidemics of obesity, and this trend of increasing disease occurrence in young women is expected to further increase. In younger women, the development of atypical endometrial hyperplasia or endometrial cancer is often associated with obesity and anovulation [5]. Further risk factors include history of oligomenorrhea, chronic anovulation and polycystic ovary syndrome (PCOS), infertility, nulliparity, and diseases related to excessive production of estrogens. In these cases, the balance between estrogens and progesterone is disturbed and relatively higher levels of estrogens lead to excessive proliferation of endometrial tissue, development of endometrial hyperplasia, and endometrial cancer.

The diagnosis of cancer is traumatizing for any woman, but the prognosis in terms of survival in young women undergoing standard surgical management is in the vast majority of cases excellent. As endometrial cancer causes symptoms of irregular bleeding, most young women are diagnosed with early stage adenocarcinoma that is most commonly low grade and rarely grows invasively into the myometrium [6,7]. In these cases, standard surgical management results in 5-year survival that exceeds 90% [8].

In addition to tumor spread, the prognostic factors in endometrial cancer include histological type, grade, and lymphovascular space invasion (LVSI). Since 1983, endometrial cancer has been classified using mostly dualistic approach, classifying tumors into type I and type II. Type I tumors are of endometrioid subtype and type II tumors are primarily of non-endometrioid histology, such as serous, clear-cell, mixed cell, dedifferentiated tumors, or carcinosarcomas that are associated with poor prognosis [9]. Before menopause, a very small proportion of women are diagnosed with type II tumors.

The loss of reproductive function and ovarian hormone production is a serious and frustrating consequence of standard oncological management. Young patients who desire fertility preservation and have good expected oncological outcomes may prefer more conservative approaches. In these cases, it is of utmost importance to carefully select the women in whom fertility sparing treatment is oncologically safe and tailor the treatment according to their needs. Undertreatment of women with endometrial cancer in an attempt to preserve fertility may translate to excessive oncological risk and progression of the disease. On the other hand, overtreatment may cause loss of fertility, lifelong side-effects, such as increased risk of cardiovascular events, or even result in secondary malignancies. Today, this decision still relies on standard histopathological assessment of endometrial biopsy and imaging methods to assess the stage of the disease (most commonly expert ultrasound or magnetic resonance imaging (MRI)) [10]. Witnessing the developments in the techniques of endometrial biopsy assessment in the last decade, there is a possibility of more precise characterization of the disease and the risk it poses to women. This review analyses the current diagnostic algorithms and management of women with endometrial cancer that are candidates for conservative management. We will discuss the potential changes in risk stratification that molecular characterization of tumors could contribute to management of young women with endometrial cancer and the effectiveness of different available progestin options.

## 2. Materials and Methods

In view of the heterogeneity of the addressed topic and the fragmented literature, a conventional systematic review was not considered appropriate. Instead, we aimed to systematically search the academic databases to produce a critical narrative overview of the current state of knowledge [11]. We searched PubMed (https://pubmed.ncbi.nlm.nih.gov), Cochrane Library (https://www.cochranelibrary.com), Scopus (https://www.scopus.com), and Web of Science (https://webofknowledge.com/WOS) for English language sources using the following keywords: endometrial cancer, fertility preservation, young women, reproductive function, and molecular classification. The publications cited in these articles were then screened and selected for review if relevant. Preference was given to the sources published in the last ten years. All types of articles were considered for review. Exclusion criteria were articles for which full text was not available or articles not in English language. The ClinicalTrials registry (https://www.clinicaltrials.gov) was searched to identify the ongoing studies of conservative endometrial cancer management in young women. The search terms used were: endometrial cancer and young women. Additionally, the guidelines of the relevant scientific societies were considered for the review.

## 3. Results

The described strategy yielded 52 papers to be included in the review (Figure 1). In addition, the recommendations from relevant societies focusing specifically on the management of young women have been collected and are presented in Table 1. Because our area of interest was represented by a heterogeneous collection of work, a conventional systematic review was not performed and consequently this review presents a narrative, thematic summary, and appraisal of this field.

### 3.1. Currently Used Diagnostic Methods in Women with Endometrial Cancer and Their Limitations

All women diagnosed with endometrial cancer should undergo comprehensive evaluation before commencing treatment, but this is even more crucial in case conservative management is considered. Diagnostics minimally consists of pre-surgical assessment of stage and grade of the disease. The routine steps in this procedure include complete medical history, endometrial sampling, and pelvic and abdominal imaging. Myometrial invasion and grade of the tumor are currently recognized as the most important prognostic factors, and the selection of women eligible for fertility sparing treatment is today largely based on these two parameters [12]. However, it should be emphasized that the final diagnosis of tumor spread and grade is based on examination of the uterus after hysterectomy. In women seeking fertility sparing options, we have to rely on imaging methods to assess the stage and endometrial biopsy specimen analysis to evaluate the grade. Therefore, these represent the cornerstone of pretreatment diagnostics. The discordance between the results of pre-surgical investigations and the final post-hysterectomy diagnosis is not uncommon; thus, especially in women deciding for conservative management, all efforts should be made to reach the most accurate estimation [15,16,17].

Myometrial and cervical tumor invasion can be assessed by transvaginal ultrasound or MRI. Although the data showing superiority of either method is controversial, latest reports suggest that the diagnostic accuracy of both modalities to diagnose myometrial invasion is comparable [10]. The dilatation and curettage (D&C) has long been considered the method of choice for endometrial sampling, but according to the very recent European Society of Gynaecological Oncology (ESGO)/European Society for Radiology and Oncology (ESTRO)/European Society for Pathology (ESP) [12] consensus, endometrial sampling should be performed by hysteroscopy. Hysteroscopy offers direct visualization of the endometrium and has been demonstrated to result in higher agreement with the final histological diagnosis compared to D&C. Nonetheless, it has been demonstrated that a certain degree of error is unavoidable [15] and women should be aware that even in well-planned and executed diagnostic procedures, under-staging can occur and women with higher stage or grade of the disease may commence with conservative management.

### 3.2. Management of Women Deciding for Fertility Sparing Approach

#### 3.2.1. Candidates for Fertility Sparing Management Based on Tumor Grade and Stage

According to the current European Society of Gynaecological Oncology (ESGO)/European Society for Radiology and Oncology (ESTRO)/European Society for Pathology (ESP) consensus [12] and The American College of Obstetricians and Gynecologists (ACOG) practice guidelines, only women with grade 1 endometrial cancer without evidence of myometrial invasion can be offered the possibility of fertility sparing management options. It is suggested that grade 2 tumors without myometrial invasion should only be counseled about fertility sparing treatment in highly experienced centers, as current evidence of safety in this setting is sparse. Only British Gynaecological Cancer Society (BGCS) guidelines consider also women with superficial myometrial invasion as potential candidates (Table 1).

Patients with grade 2 or 3 disease have significantly worse prognosis compared to grade 1 patients and this is today generally considered a contraindication for conservative management [18]. A large retrospective study has also concluded that use of progestins in women with FIGO IB disease (myometrial invasion >50%) is related to decreased survival [19]. Based on a landmark Gynecologic Oncology Group (GOG) study from the 1980s, patients with grade 1 endometrial cancer and inner third myometrial invasion still had about 3% and 1% risk of pelvic and paraaortic lymph node spread, respectively [20]. In cases where grade 1 endometrioid tumor was confined to the endometrium, there were no patients with metastatic disease and the risk of extrauterine spread was negligible [20]. In the same study, the risk of lymph node spread for grade 2 endometrial cancer was 3% (both pelvic and paraaortic) when the tumor was confined to the endometrium. Similar findings were confirmed also in subsequent studies; a very recent large-scale retrospective study focusing specifically on young women with endometrial cancer included 1284 cases and showed that the rate of lymph node metastasis in grade 1 tumors without myometrial invasion was 0.5%. With up to 50% myometrial invasion, this risk was increased to 1.6%. In grade 2 and 3 tumors, this risk was 2.1% and 4.3%, respectively [21]. In the last decade, with the widespread use of minimally invasive surgery and introduction of sentinel lymph node dissection and pathological ultra-staging, studies have shown higher rates of lymph node metastases. A prospective study including 187 patients with stage I–II endometrial cancer has shown that almost 50% of micrometastases are missed when conventional staging is performed in comparison with sentinel lymph node dissection and ultrastaging [22]. A study focusing specifically on sentinel lymph node dissection has shown that in grade 1 endometrial cancer, ultra-staging results in 1.2% of lymph node metastasis in cases without myometrial invasion and 12% in cases with myometrial invasion (even superficial) [23]. Therefore, myometrial invasion is considered to be the most important factor for lymph node metastasis and is today generally considered a contraindication for conservative management [24]. However, the clinical importance of micrometastases is still not clear today and it is not definite that women with lymph node micrometastases have worse prognosis compared to women without micrometastases [25,26]. It should be mentioned that it is difficult to estimate the influence of micrometastases on the clinical outcome, as most women with micrometasases would today receive adjuvant treatment according to current guidelines [27,28]. Nonetheless, currently available data shows that the use of sentinel lymph node dissection allows for higher sensitivity in detecting possible lymph node metastases compared to standard lymph node dissection. Therefore, it remains to be investigated in future studies whether sentinel lymph node dissection may be a tool that could be used in women to specifically select a subgroup with less aggressive tumors that could opt for fertility sparing management. This could possibly extend the indications for safe conservative management options even to certain women with superficial myometrial invasion without the evidence of lymph node involvement.

#### 3.2.2. Assessing the Risk of Synchronous Ovarian Cancer

Ovarian conservation is an important factor in improving quality of life in premenopausal women [29]. Pelvic imaging is a crucial part of diagnostics not only for determining the predicted stage of the disease, but also to detect possible synchronous ovarian tumors. The data reporting this risk is unfortunately sparse, mostly retrospective, and of low quality. An early retrospective study by Walsh et al. including 102 young women with endometrial cancer (aged <46 years) has demonstrated an incidence of coexisting ovarian malignancies in up to 25% cases [30]. Almost half of these women had well-differentiated endometrial cancer with <50% myometrial invasion. However, a subsequent large-scale retrospective study evaluating synchronous cancer in 260 young women with endometrial cancer has shown that only 21 out of 471 women (4.5%) had synchronous ovarian cancer. This risk has been shown to be even lower and appears to be <1% in case of disease macroscopically confined to the uterus [31]. The largest retrospective study including young women by Song et al [32]. included 471 women with endometrial cancer that were younger than 40 years. In this group of women, there was 4.5% risk of synchronous ovarian cancer; however, in women with low-risk disease (no myometrial invasion, grade 1 endometrioid histology, normal looking ovaries), there were no cases of ovarian cancer detected [32]. A large retrospective study, not focusing specifically on young women, has shown that women with endometrial cancer and synchronous ovarian cancer are more likely to be younger and less likely to be of black ethnicity. In women younger than 40 years, the incidence of synchronous ovarian cancer was again 4.5% [33]. This shows that although the risk of synchronous ovarian malignancy seems to be relatively low, when deciding for ovarian conservation, this possibility should be considered.

The second theoretical risk of ovarian conservation is related to continuous hormone secretion that might stimulate the growth of hormone dependent tumor cells. While this could arguably present an obstacle to ovarian conservation, it has been shown that hormone replacement therapy is safe in case of stage I and stage II endometrial cancer and at least in this setting, ovarian conservation should be possible [34]. Meta-analyses have shown that in premenopausal women with good preoperative imaging evaluation, ovarian conservation is a safe approach that does not impose a risk to oncological outcomes [35,36]. A recent, large scale retrospective study, has also shown there is a low risk of endometrial cancer ovarian metastases in women with early-stage, low-grade endometrial cancer and ovarian conservation is a safe approach in this population of women [37]. Considering the fact that routine scanning of the ovaries when performing gynecological pelvic ultrasound does not require significant additional time or cost, this procedure should be considered as part of a good clinical practice. It provides information that is crucial when deciding for fertility sparing options, but even in women undergoing surgical management, this allows for better management plan and the possibility of ovarian conservation. In women with high-risk endometrial cancer, when ovarian conservation is not considered to be safe, the possibility of ovarian stimulation with oocyte cryopreservation for possible surrogacy could also be considered. In these difficult cases, extensive counseling should be provided to understand the oncological outcomes and the realistic possibilities of future fertility treatment.

#### 3.2.3. Choice of the Best Regimen for Conservative Management of Endometrial Cancer

According to the current recommendations [12], women considering fertility sparing treatment should be counseled that this represents a deviation from the recommended standard of care and is supported by limited evidence. They should be offered close surveillance after successful regression. The recurrence after progestin treatment was shown to be time-dependent and between 30–40% of women are likely to experience recurrence [38,39]. This means that patients are generally advised to undergo hysterectomy after finished childbearing [40,41]. A survey of European clinicians managing women with endometrial cancer has shown that although most believe women with grade 1 endometrial cancer without myometrial invasion are candidates for fertility sparing management, few women are actually managed conservatively. There was no consensus whether progesterone receptor status should be considered and if patients with Lynch syndrome could be considered as appropriate candidates [42]. This indicates the need for better prognostic markers.

Conservative medical management is currently based on progestins with medroxyprogesterone acetate (MPA; 400–600 mg/day) or megestrol acetate (MA; 160–320 mg/day) for at least six months. Meta-analysis including young women with early stage endometrial cancer has shown that a complete response of treatment occurs in about 80% of patients, and the plateau of response occurs after 12 months of progestin treatment. Recurrence occurred in 17% after 12 months and in 29% after 24 months after treatment [38]. Long-term oral administration of large doses of progestins is associated with adverse gastrointestinal side effects, which restricts its use in certain populations and limits compliance in young women. Therefore, studies have compared the use of levonorgestrel-releasing intrauterine system or oral letrozole combined with gonadotropin-releasing hormone agonist therapy to standard oral therapy. A recent study showed the overall response rate with levonorgestrel-releasing intrauterine system therapy alone was successful in 75% at 6 months [43]. However, there is skepticism on using levonorgestrel-releasing intrauterine system alone, as a recent meta-analysis demonstrated that patients with endometrial cancer/atypical endometrial hyperplasia who are treated with progestins, with or without levonorgestrel-releasing intrauterine system, and levonorgestrel-releasing intrauterine system alone can reach good response rates, but levonorgestrel-releasing intrauterine system group without any additional treatment had the worst pregnancy outcomes [44]. However, the data are of low quality and further studies are needed to confirm these findings. One Korean study reported high success rates with 87.5% of patients reaching complete remission after 9.8 months after using oral progestins in combination with levonorgestrel-releasing intrauterine system [45]. Levonorgestrel-releasing intrauterine system combined with gonadotrophin-releasing hormone agonists had a success rate of 75.5% in achieving a complete response, according to meta-analysis, published by Fan et al. [46].

Progestin therapy has also been studied in combination with hysteroscopic tumor resection [47]. There is concern for developing intrauterine adhesions, and the possibility of peritoneal cancer spread after hysteroscopy, but none of these risks have been confirmed in prospective studies [48]. Although only a limited number of patients undergoing hysteroscopic resection followed by progestin treatment were included in a recent meta-analysis (*n* = 73), the success rates in terms of complete remission appear to be higher (95.3%) compared to progestin treatment alone (72.9% in oral progestins and 76.3% in intrauterine progestins). The pregnancy rate was similar among the three groups (47.8–56.0%) and the risk for recurrence appeared to be higher in the group using oral progestins [46].

A few reports [49,50] evaluated fertility sparing treatment in women with superficial myometrial invasion. Casadio et al. [49] reported on a group of 82 women opting for fertility sparing procedures in endometrial cancer and atypical endometrial hyperplasia. Among them, in the grade 1 endometrial cancer group, 5/36 women (13.8%) had superficial myometrial invasion. In these women, hysteroscopic resection and progestin therapy were attempted. In a 60-month follow-up period, recurrence was observed in four out of 36 women. In the subgroup of five women with superficial myometrial invasion, the cancer recurred in one woman after the end of hormonal treatment. In two women with initial superficial myometrial invasion, disease recurred after 60 months of follow-up as atypical endometrial hyperplasia. All these women were then treated with total hysterectomy and histologic diagnosis was confirmed after surgery.

Discovery of metformin’s antiproliferative action on endometrial cancer cells has prompted its use in women with endometrial cancer. In vitro studies have demonstrated that metformin suppresses endometrial cancer cell growth leading to autophagy and apoptosis [51]. Although the quality of evidence was assessed as low and the heterogeneity of studies in this area is significant, a systematic review has concluded that addition of metformin may improve regression of atypical endometrial hyperplasia to normal and reduce proliferation markers implicated in tumor progression in patients with advanced endometrial cancer [52]. A recent retrospective study including women with atypical complex endometrial hyperplasia has shown that metformin may be of benefit in women using progestins in the form of levonorgestrel-releasing intrauterine system [53]. In contrast to these findings, a very recent retrospective study focusing specifically on young women seeking fertility sparing treatment concluded that addition of metformin to progestins does not improve remission rates [54]. There are multiple ongoing studies investigating the addition of metformin to different forms of progestins in women seeking fertility sparing management (Table 2) and the value of metformin in this setting remains to be established.

### 3.3. Incorporating Tumor Biology into Management Algorithms

The most common classical risk factor for endometrial cancer is considered to be estrogen excess. In the last decade, we have witnessed a growing interest in the identification of other risk factors and stratification of women based on the biology of the disease. A retrospective study focusing on younger women with endometrial cancer (under 50 years of age) has shown that based on the causes and the interconnected biology, women with endometrial cancer can be classified into three groups. These are (i) “high estrogen” group (57.7% cases), (ii) the “Lynch” group (8.3% cases), and (iii) the “neither” group (34.2% cases) [57]. This is especially important in patients with progestin therapy failure, as in these patients, there can be higher influence of other, progestin-independent signaling pathways [58].

The presence of Lynch syndrome in young women with endometrial cancer is higher compared to the general population [59]. Inherited genetic mutations predisposing women for endometrial cancer account in the general population for 3–5% of cases. The most prevalent genetic mutation predisposing women to endometrial cancer among inherited genetic mutations is Lynch syndrome (present in 2–3% of all patients) [60]. Prospective trials testing women with endometrial cancer under the age of 50, however, show that in this population, Lynch syndrome was present in 9% of cases [59]. A recent data analysis added that in women under 50 years of age, microsatellite instability (marker for further testing for Lynch syndrome) was highly expressed in 11% of stage IA tumors. It was also shown that endometrial cancer characterized by microsatellite instability is more frequently present in advanced stages and is associated with lymphovascular space invasion more frequently than sporadic cancer [61]. Currently, there is no unified consensus on guiding testing for Lynch syndrome in women opting for fertility sparing management. A web-based study of the European Network of Young Gynecologic oncologists (ENYGO) showed that patients with endometrial cancer and Lynch syndrome were considered as candidates for conservative management by nearly half of the survey participants (49.1%), but nearly half also disagreed (47.5%) [42]. Undergoing fertility sparing management may limit the detection of this mutation that predisposes women to increased risk of colorectal, endometrial, and ovarian cancer. Young women with endometrial cancer should therefore always be carefully consulted about the need for Lynch syndrome testing. The possibility of conservative management in the presence of Lynch syndrome is questionable and extensive counseling is needed in these cases.

Histopathological examination of endometrial biopsy is associated with significant inter- and intra-observer variability and is unreliable in predicting the risk of disease recurrence [16,17]. Thus, novel molecular classifications to improve risk stratification of patients are being investigated today. Four distinct molecular subtypes of endometrial cancer have been identified from the Cancer Genome Atlas (TCGA), leading to several studies assessing the impact of endometrial cancer molecular classification on patient outcomes. By using genetic and surrogate immunohistochemical markers, a classification of key groups was made. These groups are POLE ultramutated (POLEmut), MMR deficient (MMRd), non-specific molecular profile (NSMP), and p53 abnormal (p53abn) endometrial cancer [62]. An early study by the TransPORTEC group has demonstrated that based on these classifiers, a group of MMR-deficient endometrial cancer cases, which is a marker of microsatellite instability (MSI), and POLEmut cases were associated with favorable prognosis. The 5-year recurrence-free survival was 93% for MMR-deficient and 95% for POLE-mutant in comparison to the p53abn group (recurrence free survival 50%) and non-specific molecular profile (NSMP) subgroup where recurrence free survival was 52% [63]. Further research showed that POLEmut endometrial cancer patients had a 10-year recurrence free survival of 100% in comparison to the POLE wild type group which had a recurrence free survival of 80.1% [64]. Another study showing prognostic applicability of molecular classification was the Proactive Molecular Risk Classifier for Endometrial Cancers (ProMisE) study [65]. In a cohort of 257 young patients (below the age of 50 years) it showed a benefit towards overall survival and recurrence free survival in the POLEmut and p53 wild type group. Overall and recurrence-free survival was significantly lower in p53 abnormal and MMR deficient patient subgroups. The molecular profile of patients in this study showed, that young women with endometrial cancer were classified as 19% MMR immunohistochemically abnormal, 13% POLE exonuclease domain mutations (EDM), 4% p53 abnormal, and 64% p53 wild type [65]. Adding to that is data of a small cohort (*n* = 15) analyzed by Falcone et al. [66], reporting in their analysis of biopsy samples of young women with endometrial cancer, that 46.7% were MMR abnormal, 6.6% POLE EDM, 0% p53 abnormal and 46.7% p53 wild type [66]. A recent review of low-grade endometrial cancer molecular profiles showed that certain molecular subtypes of (i) POLE-ultramutated tumors; (ii) p53 wild type/NSMP tumors with wild type CCND1 and CTNNB1, are estrogen receptor (ER) and progesterone receptor (PR) positive and lack 1q32.1 amplification, and with low L1CAM expression and DNA damage; and (iii) MMR-deficient tumors with wild type CCND1 are estrogen receptor and progesterone receptor positive, lack PTEN methylation, and, without Lynch syndrome, could be potentially considered low-risk endometrial cancer tumors [67]. This is in contrast to the earlier findings that MMR-deficient type cases could be considered as “low-risk” and benefit from progestin therapy. Therefore, further refinements are needed to improve the molecular characterization of these women. This indicates the need for further investigations of additional molecular markers, such as mutational status of beta-catenin (CTNNB1), L1 cell adhesion molecule (L1CAM), estrogen receptor and progesterone receptor status, PTEN methylation status and CCND1, and other markers that could potentially improve the risk stratification models in the future. Molecular markers can also be combined with immunohistochemical investigations on tumor biopsy samples before planned treatment and clinical parameters to create a reliable prediction model [67]. Travaglino et al. [68] have recently published a systematic review on immunohistochemical markers predicting response to treatment and evaluating treatment during follow-up period. The analysis (depicted in Figure 2) revealed markers which belong to different pathways potentially involved in the good response or resistance to progestin therapy [68].

The study showed that staining for progesterone and estrogen receptor expression was important prior to treatment, but that an indicator of good response was also Dusp6, a marker of the mitogen-activated protein kinase (MAPK) signaling pathway. An absent expression, deficiency of MMR, was an important sign of potential therapy failure as was the involvement of PTEN, an activator of the mTOR/AKT/PI3K pathway. This shows that next to understanding of progesterone and estrogen signaling, further focus is needed towards the pathways of MAPK and mTOR/AKT/PI3K.

A recent study including 81 women with endometrial cancer has shown that the incidence of CTNNB1 somatic gene mutation was higher in early onset endometrial cancer and the immunohistochemical accumulation of beta-catenin was inversely correlated with patients’ age [69]. It has been proposed that overexpression of beta-catenin could be an of treatment failure in fertility sparing management [58]. A growing body of indicator evidence indicates the clinical importance of CTNNB1 mutation. Especially in young patients, there are propositions of establishing CTNNB1 status as the 5th molecular classifier of endometrial cancer [62]. CTNNB1 evaluation in a case-control study showed that the odds of disease recurrence, if the CTNNB1 mutation was present, were nine times higher than in those the mutation was not present [70]. Kurnit et al. [71] evaluated the molecular expression in presumably low-risk endometrial cancer and found that in stage I or II, the presence of either Tp53 mutations or CTNNB1 mutations represented a risk factor for disease recurrence in otherwise presumed low-risk patients [71]. On the other hand, a significant discordance between immunohistochemical expression of beta-catenin and CTNNB1 gene profiling status has been demonstrated and further studies are needed to support its clinical value [62]. While Tp53 has relatively low copy number alterations, further proposed candidates for risk stratification can be found in frequently mutated genes, such as PTEN, PIK3CA, ARID1A, and KRAS [72]. Representatives of the PTEN and PI3K/Akt/mTOR signaling pathway were also associated with poor outcomes [60,68], but the association was only significant when models for risk stratification incorporated PTEN status and one of the molecular marker statuses [60,68]. In silico analysis reported novel gene signatures which were associated with a worse risk profile in endometrial cancer. These were CDK1, KIF2C, UBE2C, and TPX2. These genes were associated with worse overall survival in endometrial cancer. Validated on tissue samples, TPX2 was shown to be relevant for risk assessment especially in grade 1 and 2 endometrial cancer [73]. Further bioinformatic research with archival sample validation also proposed, that UBE2C could serve as a marker of worse prognosis [74]. Figure 3 depicts the candidate genes adding to the pool of risk stratification markers. The depiction of different markers in Figure 2 and Figure 3 shows, that there is still much unknown about the interactions of different molecular signaling pathways [68,73,74,75]. The major pathways involved in endometrial cancer seem to be PI3K/Akt/mTOR, MAPK, WNT, and FBXW7 signaling. While in isolation several studies have shown that these markers have potential predicting abilities in endometrial cancer, they have yet to be understood in terms of the Cancer Genome Atlas (TCGA) risk profiles. Currently, we do not understand how a specific molecular profile of the tumor is connected to the signaling pathways that are important in endometrial cancer.

### 3.4. Implications of Molecular Markers for the Future of Fertility Sparing Management

There are many future challenges in improving risk stratification for fertility sparing management. In patients currently classified as low-risk early stage according to established recommendations, there are still patients with increased risk of disease recurrence and spread. This is supported by recent reports showing recurrence rates in well differentiated stage IA cancers, where in a cohort of 2691 women in a median follow-up time of 6.1 years, the incidence of recurrences was 7.2% [76]. The initial studies that are available show that endometrial cancer molecular classification and different molecular markers further refine the patient risk profiles and not all stage IA, grade 1 patients bear the same disease biology. As Bosse et al. [77] showed by evaluating endometrioid cancer samples that are grade 3 and generally deemed to have a higher risk of recurrence, their biological risk profile differs greatly if molecular classification is applied to them. It was shown, that the currently grade 3 tumor group (*n* = 381) was assembled of 12.9% of POLE ultramutated tumors, 20.7% of p53 abnormal tumors, 30.2% non-specific molecular profile (NSMP) tumors, and 36.2% of MMR deficient tumors. Further analysis of recurrence-free survival showed that POLE ultramutated and MMR-deficient remained independent prognostic factors for improved recurrence free survival in the grade 3 endometrioid cancer group [77]. Although the molecular classifiers are promising for the future management of women, currently available data do not allow for clinical applicability to extending the potential candidates for fertility sparing management. As new genetic signatures are identified through the transcriptional endometrial cancer landscape, finding a signature that could be incorporated into a prognostic model combined with the other molecular identifiers will be needed.

## 4. Discussion

Management of endometrial cancer in young women and selection of women that are candidates for fertility sparing conservative treatment are challenges that are becoming increasingly complex. According to the current recommendations, this is today largely based on classical histopathological specimen evaluation and pelvic imaging. The promising investigations of novel endometrial cancer biomarkers are likely to change this approach in the future; however, there is currently insufficient data to support their routine use in this clinical setting.

With increasing incidence of endometrial cancer in younger women, increasingly more women are likely to seek conservative management options. Today, there is considerable data showing that in low-risk endometrial cancer, without adnexal metastases seen by MRI or ultrasound, or hereditable conditions (Lynch syndrome), ovarian conservation should be considered safe in women before menopause. This change in surgical management can have a significant impact on the quality of women’s life. The most significant challenge in this setting is to balance the potential benefits of sparing fertility or conserving the ovaries to minimizing the risk of negative oncological outcome. Therefore, clinicians and patients need better and reliable tools to plan their management. This could be achieved by better quality imaging modalities, improved histopathological assessment, and molecular characterization of tumors. In any case, the prognosis after fertility sparing management should not be worse compared to primary radical surgical treatment.

The techniques of ultrasound imaging as well as MRI have improved significantly over the last decade. This allows for more accurate assessment of tumor size, vascularity and myometrial or cervical stromal invasion. Although most of these markers are currently of limited value, further studies evaluating whether detailed, expert ultrasound scan analyzing these markers can distinguish low-risk endometrial cancer from more high-risk variants are warranted [78]. It also remains to be evaluated in the future trials, what is the risk of lymph node metastasis by sentinel lymph node mapping in women with well-differentiated tumors and superficial myometrial invasion. These cases are today considered controversial for conservative management, but a subgroup of women could possibly benefit from fertility sparing treatment and delaying hysterectomy.

Molecular characterization of endometrial cancer is changing the approach to diagnosis and management of women. It is likely that certain subgroups of POLE-ultramutated tumors, p53 wild type/non-specific molecular profile (NSMP) tumors and MMR-deficient tumors with specific low risk characteristics could potentially be considered as low-risk for advancement of disease and be possible candidates for fertility sparing management. However, the currently available data suggest that addition of other predictive markers to the model, such as determination of CCND1 and CTNNB1 status and progesterone and estrogen receptor expression will improve the predictive ability of these markers. Universal molecular tumor screening can also increase the detection rate of Lynch syndrome in young women. This is the group of women that is also probably at higher risk of disease progression and development of synchronous cancer, and needs especially careful counseling about the possible benefits and risks before deciding for conservative approach. Unfortunately, there is currently no sound evidence on the safety of fertility sparing approach in this population of women. Considering that 20–70% of women carrying a mutation in one of the MMR genes will develop endometrial cancer [79], appropriate counseling is needed. It has been shown that women carriers of pathological MMR gene variants (that is, MHL1, MSH2, and MSH6) had a cumulative risk to develop additional ovarian cancer (at age 75 years) of up to 10–17% [80]. These data need to be considered when counseling and shared decision-making on fertility sparing approaches in women with Lynch syndrome. In the future, when deciding for fertility sparing management, immunohistochemical and clinical parameters described in our review will probably also need to be included in decision algorithms to reach a satisfactory predictive model.

## 5. Conclusions

In conclusion, the prognosis for early-stage endometrial cancer has been shown to be excellent regardless of age. Most women diagnosed with early-stage endometrial cancer are more likely to die from cardiovascular disease than from cancer. With the advancements in molecular characterization of the disease, there is more than ever a need and the possibility to develop better risk stratification models to guide treatment in young women. One approach fits all model should be replaced with patient focused management. Further studies in this setting are warranted to make a step forward in clinical care and improve quality of life in this group of women.

## Figures and Tables

**Figure 1 cancers-13-00602-f001:**
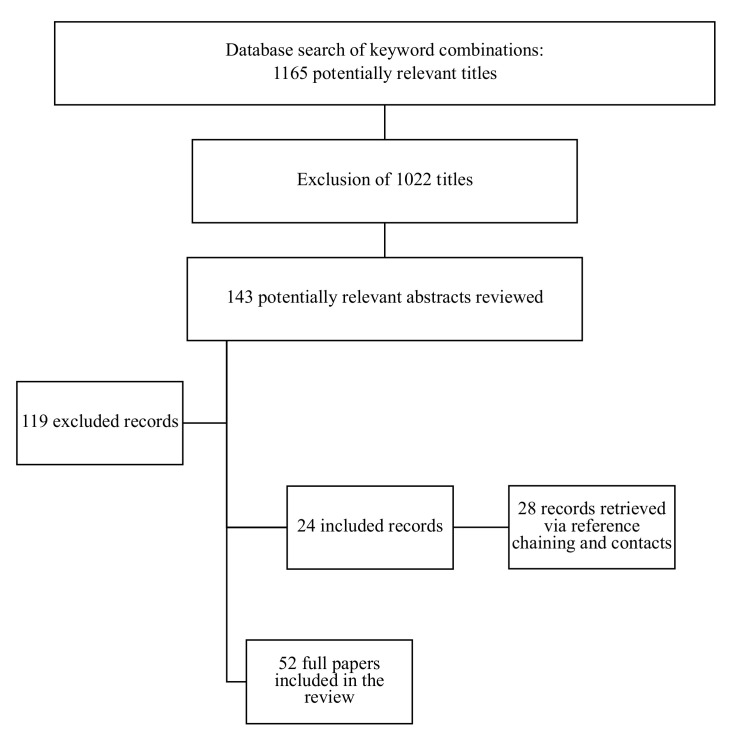
Study selection process.

**Figure 2 cancers-13-00602-f002:**
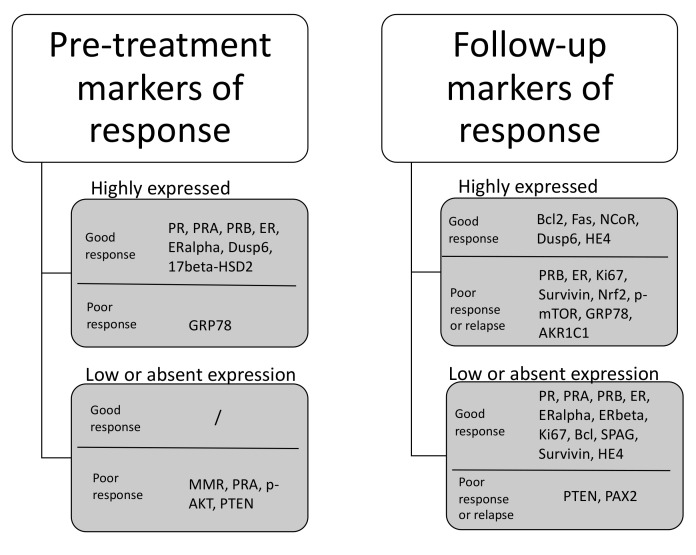
Summary of statistically significant immunohistochemical markers for conservative management [68].

**Figure 3 cancers-13-00602-f003:**
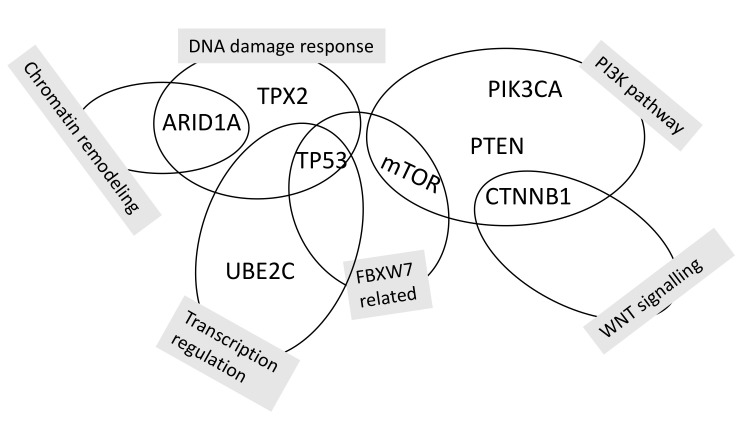
Functional grouping of candidate genes for potential risk stratification in early endometrial cancer [68,73,74,75].

**Table 1 cancers-13-00602-t001:** Scientific societies’ guidelines on selection of women with endometrial cancer that are candidates for fertility sparing management.

Scientific Society	Selection Criteria for Fertility Sparing Treatment			
	Histology	Tumor Stage	Specimen Obtainment Method and Recommended Imaging Modality	Other Recommendations
ESGO-ESTRO-ESP Consensus Conference on Endometrial Cancer [12]	Well-differentiated (grade 1) endometrial adenocarcinoma or premalignant state (atypical hyperplasia).	Tumor confined to the endometrium. No extrauterine involvement (adnexal or pelvic nodes).	Histology and grade confirmed by hysteroscopy by an expert pathologist. Myometrial invasion or adnexal involvement excluded by MRI. Expert ultrasound can be considered as an alternative.	Women should understand the non-standard nature of conservative treatment and the need for close follow-up. Women must be informed of the need for future hysterectomy.
The American College of Obstetricians and Gynecologists (ACOG) and Society of Gynecologic Oncology (SGO) Clinical management guidelines [13]	A well-differentiated, grade 1, endometrioid endometrial carcinoma.	No myometrial invasion. No extrauterine involvement (no synchronous ovarian tumor or metastases, no suspicious retroperitoneal nodes).	Dilatation and curettage may be better than office endometrial biopsy. MRI may be the preferred modality compared to ultrasonography and CT to evaluate the presence of myometrial invasion.	Strong desire for fertility sparing. No contraindications for medical management. Patient understands and accepts that data on cancer-related and pregnancy-related outcomes are limited (informed consent).
British Gynaecological Cancer Society (BGCS) Uterine Cancer guidelines [14]	Selected women with grade 1 endometrial cancer.	No invasion or superficial myometrial invasion.	MRI imaging to exclude >50% myometrial invasion, adnexal or nodal involvement.	Specialist gynae-pathology multidisciplinary team review is required. Women should be counseled carefully about the current known response rates on progestins and progression risk. Women should be offered genetic counseling to exclude the presence of Lynch syndrome.

**Table 2 cancers-13-00602-t002:** Currently ongoing studies of different conservative management options in women with endometrial cancer.

CT Identifier	Recruitment Criteria	Response Markers	Intervention	Status	Primary Completion
NCT03804463	Women with EC up to 50 years of age, with EC Stage IA G1 or G2, normal CA125	Histology and recurrence free survival	radical surgery vs. endometrectomy vs. histerectomy with ovarian preservation	Not yet recruiting	January 2019
NCT03538704	EC up to 40 years of age, Stage IA G1 or G2	Complete pathological response and fertility, RFS	MPA or MPA + metformin	Recruiting	March 2020
NCT02990728	EC from 20–40 years of age, Stage IA G1, no imaging signs of adenomyosis or ovarian endometriosis, normal CA125	Histology and markers: progesteron A and B receptors, estrogen, Ki7, PTEN, Bcl2	Mirena vs. Mirena + Metformin	Unknown	March 2018
NCT03567655	EC from 20–40 years of age, Stage IA G1 or G2 with superficial invasion	Histology and recurrence free survival	MPA	Not yet recruiting	October 2022
NCT04362046	EC from 19–39 years of age and Stage IA G1, up to 1/3 of myometrial invasion and AEH	Local/distant disease and fertility outcomes	6 months of high-dose progestin and then hysteroscopic resection	Not yet recruiting	May 2026
NCT04008563	EC from 18–41 years of age and Stage I, G1 or AEH and BMI ≥ 35, no signs of disease beyond uterus	Patient reported outcomes and complete response rate vs. LNG-IUS alone	Mirena and bariatric surgery vs. Mirena	Not yet recruiting	August 2022
NCT04046185	EC from 18–45 years of age and early EC	Histology and fertility outcomes	PD-1 inhibitor (toripalimab) every 3 weeks and megestrol acetate vs. megestrol acetate alone	Not yet recruiting	October 2022
NCT03463252	EC up to 40 years of age, Stage IA, G1 with positive progesterone receptors or AEH, no Lynch syndrome	Histology and fertility outcomes	randomized study of MPA only or MPA + LNG-IUS or GnRH-a + LNG-IUS or LNG-IUS only for EC or AEH	Recruiting	December 2019
NCT01594879 [55]	EC up to 40 years of age, Stage IA, G1without myometrial invasion	Treatment response rate and diagnostic accuracy	Combined oral MPA/LNG-IUS treatment and evaluation of hysteroscopy vs. D&C for follow up	Unknown	December 2014
NCT01686126 [56]	EC from 18 years of age wishing to retain fertility or unfit candidate for surgery due to comorbidity with Stage IA G1 EC or AEH, normal CA125	Complete histological response, secondary: serum and molecular markers	LNG-IUS + Metformin, LNG-IUS + weight loss, LNG-IUS alone	Active, not recruiting	August 2020
NCT03241914	Stage I EC with no deep myometrial invasion in patients between 18 and 45 years of age	Pathological response rate	Megestrol acetate alone vs. megestrol acetate and LNG-IUS	Recruiting	July 2020
NCT00788671	Stage IA EC, G1 or AEH with fertility preservation wish or comorbidities preventing safe surgery	Pathological response rate, IHC, estrogen and Wnt signaling response	LNG-IUS placement	Active, not recruiting	November 2020
NCT02035787	Stage IA EC, G1 or AEH with fertility preservation wish or comorbidities preventing safe surgery	Pathological response	LNG-IUS + Metformin	Recruiting	December 2022
NCT02397083	Pre- and post-menopausal women with Stage IA EC, G1 or AEH	Pathological response	LNG-IUS alone or LNG-IUS with mTOR inhibitor everolimus	Recruiting	September 2026

Information valid as of 25 October 2020 based on ClinicalTrials.gov registry.

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
