# Peer review of "The Perspectives of Fertility Preservation in Women with Endometrial Cancer"

_cancers, 2021, doi:10.3390/cancers13040602_

Round 1

Reviewer 1 Report

I had the pleasure to review this well written manuscript for Cancers. This is an interesting review about fertility sparing treatments in women affected by endometrial cancer. The topic is attractive and the focus on molecular characterization of endometrial cancer is valuable. However, additional clarifications are needed in order to better interpret the findings of this study. Moreover, a spell and syntax check is required.

ABSTRACT

Please restate line 26-27. The syntax of the sentence is not correct.

INTRODUCTION

Introduction seems to be too long with several unnecessary sentences for this section. In my opinion you could move them to the discussion section (e.g. Line 66-73; Line 77-83)

RESULTS

Line 200: you reported that endometrial sampling should be performed by D&C and not by Pipelle or hystreroscopy. In my opinion Literature about this issue is not so clear (Epstein et al. Dilatation and curettage fails to detect most focal lesions in the uterine cavity in women with postmenopausal bleeding. Acta Obstet Gynecol Scand 2001; 80:1131. Gimpelson RJ, Rappold HO. A comparative study between panoramic hysteroscopy with directed biopsies and dilatation and curettage. A review of 276 cases. Am J Obstet Gynecol 1988; 158:489.). Several studies showed that hysteroscopy can detect focal lesions of the endometrium that could be missed by D&C alone. Moreover, endometrial sampling under direct visualization of the uterine cavity seems to be an obvious benefit for the study of the uterine cavity. Please discuss.

I suggest incorporating lines 211-247 in a separate paragraph about Synchronous ovarian tumors.

Line 250: Myometrial invasion is considered one of the most important factors for the risk of recurrence and lymph nodes metastasis. However, several studies have started to evaluate outcomes for young women affected by endometrial cancer and treated by fertility sparing approach (Casadio et al. Fertility Sparing Treatment of Endometrial Cancer with and without Initial Infiltration of Myometrium: A Single Center Experience. Cancers (Basel). 2020 Nov 29;12(12):3571. doi: 10.3390/cancers12123571; Casadio et al. Fertility-Sparing Treatment of Endometrial Cancer with Initial Infiltration of Myometrium by Resectoscopic Surgery: A Pilot Study. Oncologist. 2018 Apr;23(4):478-480. doi: 10.1634/theoncologist.2017-0285). Please discuss them.

Author Response

We are very grateful to the reviewer for their thoughtful and encouraging comments. We have considered all the suggestions and revised the manuscript accordingly. Please find our specific responses below.

  1. ABSTRACT Please restate line 26-27. The syntax of the sentence is not correct.

The lines 26-27 have been rewritten as follows »More focus needs to be aimed towards the preservation of quality of life without jeopardizing oncological outcomes.«

  1. INTRODUCTION: Introduction seems to be too long with several unnecessary sentences for this section. In my opinion you could move them to the discussion section (e.g. Line 66-73; Line 77-83)

The introduction lines 77-83 have been incorporated in the new paragraph on synchronous ovarian tumors (lines 362 – 407). We have also shortened the introduction, also the senctence at lines 66-73 was considered and adjusted appropriately.

  1. RESULTS: Line 200: you reported that endometrial sampling should be performed by D&C and not by Pipelle or hystreroscopy. In my opinion Literature about this issue is not so clear (Epstein et al. Dilatation and curettage fails to detect most focal lesions in the uterine cavity in women with postmenopausal bleeding. Acta Obstet Gynecol Scand 2001; 80:1131. Gimpelson RJ, Rappold HO. A comparative study between panoramic hysteroscopy with directed biopsies and dilatation and curettage. A review of 276 cases. Am J Obstet Gynecol 1988; 158:489.). Several studies showed that hysteroscopy can detect focal lesions of the endometrium that could be missed by D&C alone. Moreover, endometrial sampling under direct visualization of the uterine cavity seems to be an obvious benefit for the study of the uterine cavity. Please discuss.

We thank the reviewer for this important remark. This is indeed an issue and just at the time of this manuscript review, the new ESGO/ESTRO/ESP consensus on endometrial cancer has been published, which is an update of the 2016 ESMO/ESGO/ESTRO recommendations. We have updated this section to include the new recommendations, also showing the advantages of hysteroscopy over D&C (Lines 196-202):

»The dilatation and curettage (D&C) has long been considered the method of choice for endometrial sampling, but according to the very recent European Society of Gynaecological Oncology (ESGO) / European Society for Radiology and Oncology (ESTRO) / European Society for Pathology (ESP) consensus [18], endometrial sampling should be performed by hysteroscopy. Hysteroscopy has been demonstrated to result in higher agreement with the final histological diagnosis compared to D&C. « Table 2 was also adjusted to reflect these novel recommendations.

  1. I suggest incorporating lines 211-247 in a separate paragraph about Synchronous ovarian tumors.

We have incorporated the text on synchronous ovarian tumors in a new paragraph on synchronous ovarian tumors (lines 363 – 408).

  1. Line 250: Myometrial invasion is considered one of the most important factors for the risk of recurrence and lymph nodes metastasis. However, several studies have started to evaluate outcomes for young women affected by endometrial cancer and treated by fertility sparing approach (Casadio et al. Fertility Sparing Treatment of Endometrial Cancer with and without Initial Infiltration of Myometrium: A Single Center Experience. Cancers (Basel). 2020 Nov 29;12(12):3571. doi: 10.3390/cancers12123571; Casadio et al. Fertility-Sparing Treatment of Endometrial Cancer with Initial Infiltration of Myometrium by Resectoscopic Surgery: A Pilot Study. Oncologist. 2018 Apr;23(4):478-480. doi: 10.1634/theoncologist.2017-0285). Please discuss them.

We thank the reviewer for highlighting this dillema. We have incorporated the data on attempts of fertility sparing in endometrial cancer with superficial myometrial invasion in lines 462 – 472:

»A few reports [49,50] evaluated fertility sparing treatment in women with superficial myometrial invasion. Casadio et al. [49] reported on a group of 82 women opting for fertility sparing procedures in endometrial cancer and atypical endometrial hyperplasia. Among them, in the grade 1 endometrial cancer group, 5/36 women (13.8 %) had superficial myometrial invasion. In these women, hysteroscopic resection and progestin therapy were attempted. In a 60 month follow up period, recurrence was observed in 4 out of 36 women. In the subgroup of 5 women with superficial myometrial invasion, the cancer recurred in 1 woman after the end of hormonal treatment. In two women with initial superficial myometrial invasion, disease recurred after 60 months of follow up as atypical endometrial hyperplasia. All these women were then treated with total hysterectomy and histologic diagnosis was confirmed after surgery.«

Reviewer 2 Report

The review manuscript entitled, “The perspectives of fertility preservation in women with endometrial cancer” describes new evaluation for fertility preservation in women with endometrial cancer. The manuscript is well-written and very persuasive. However, the authors should consider the following points;

Pathogenic variant carriers of  MMR genes are elevated the risk for ovarian cancer in addition to endometrial cancers. The reviewer recommends to describe the fertility preservation in women with endometrial cancer among Lynch syndrome in the view of at risk for ovarian cancer.

Author Response

We are very grateful to the reviewer for their thoughtful comments. We have considered the suggestions and revised the manuscript accordingly. Please find our specific response below.

  1. Pathogenic variant carriers of  MMR genes are elevated the risk for ovarian cancer in addition to endometrial cancers. The reviewer recommends to describe the fertility preservation in women with endometrial cancer among Lynch syndrome in the view of at risk for ovarian cancer.

We thank the reviewer for this comment. This remark has been addressed in the lines 720 – 725, where we discuss the need to counsel patients on the risk of ovarian cancer occurance in patients with pathological variants of mismatch repair proteins:

»Considering that 20-70% of women carrying a mutation in one of the MMR genes will develop endometrial cancer, [79] appropriate counseling is needed. It has been shown that women carriers of pathological MMR gene variants (that is_MHL1, MSH2 and MSH6) had a cumulative risk to develop additional ovarian cancer (at age 75 years) of up to 10-17% [80]. This data needs to be considered when counseling and shared decision making on fertility sparing approaches in women with Lynch syndrome. «